# Maintenance of Potent Cellular and Humoral Immune Responses in Long-Term Hemodialysis Patients after 1273-mRNA SARS-CoV-2 Vaccination

**DOI:** 10.3390/ph16040574

**Published:** 2023-04-11

**Authors:** Maria Gonzalez-Perez, Jana Baranda, Marcos J. Berges-Buxeda, Patricia Conde, Mayte Pérez-Olmeda, Daniel Lozano-Ojalvo, Carmen Cámara, Maria del Rosario Llópez-Carratalá, Emilio Gonzalez-Parra, Pilar Portolés, Alberto Ortiz, Jose Portoles, Jordi Ochando

**Affiliations:** 1Centro Nacional de Microbiología, Instituto de Salud Carlos III, 28220 Madrid, Spain; 2Department of Pharmaceutical and Health Sciences, CEU San Pablo University, 28668 Madrid, Spain; 3Centro de Investigación Biomédica en Red de Enfermedades Infecciosas (CIBERINFEC), 28029 Madrid, Spain; 4Precision Immunology Institute, Icahn School of Medicine at Mount Sinai, New York, NY 10029, USA; 5Department of Immunology, Hospital La Paz, 28046 Madrid, Spain; 6Department of Nephrology, IDIPHIM Hospital Puerta de Hierro, 28220 Madrid, Spain; 7Department of Nephrology IIS-Fundación Jimenez Díaz, 28040 Madrid, Spain; 8Presidencia, Consejo Superior de Investigaciones Científicas (CSIC), 28006 Madrid, Spain; 9Department of Oncological Sciences, Icahn School of Medicine at Mount Sinai, New York, NY 10029, USA

**Keywords:** COVID-19, chronic kidney disease, hemodialysis, SARS-CoV-2 vaccine, 1273-mRNA vaccine, humoral response, cellular response

## Abstract

Continuous evaluation of the coronavirus disease 2019 (COVID-19) vaccine effectiveness in hemodialysis (HD) patients is critical in this immunocompromised patient group with higher mortality rates due to severe acute respiratory syndrome coronavirus 2 (SARS-CoV-2) infection. The response towards vaccination in HD patients has been studied weeks after their first and second SARS-CoV-2 vaccination dose administration, but no further studies have been developed in a long-term manner, especially including both the humoral and cellular immune response. Longitudinal studies that monitor the immune response to COVID-19 vaccination in individuals undergoing HD are therefore necessary to prioritize vaccination strategies and minimize the pathogenic effects of SARS-CoV-2 in this high-risk group of patients. We followed up HD patients and healthy volunteers (HV) and monitored their humoral and cellular immune response three months after the second (V2+3M) and after the third vaccination dose (V3+3M), taking into consideration previous COVID-19 infections. Our cellular immunity results show that, while HD patients and HV individuals secrete comparable levels of IFN-γ and IL-2 in ex vivo stimulated whole blood at V2+3M in both naïve and COVID-19-recovered individuals, HD patients secrete higher levels of IFN-γ and IL-2 than HV at V3+3M. This is mainly due to a decay in the cellular immune response in HV individuals after the third dose. In contrast, our humoral immunity results show similar IgG binding antibody units (BAU) between HD patients and HV individuals at V3+3M, independently of their previous infection status. Overall, our results indicate that HD patients maintain strong cellular and humoral immune responses after repeated 1273-mRNA SARS-CoV-2 vaccinations over time. The data also highlights significant differences between cellular and humoral immunity after SARS-CoV-2 vaccination, which emphasizes the importance of monitoring both arms of the immune response in the immunocompromised population.

## 1. Introduction

End-Stage Renal Disease (ESRD) patients undergoing hemodialysis (HD) are considered immunocompromised due to their vulnerability to severe acute respiratory syndrome coronavirus 2 (SARS-CoV-2) infection and their increased risk of COVID-19 mortality [1]. Vaccine effectiveness against SARS-CoV-2 is crucial for the protection of HD patients [2], especially after the SARS-CoV-2 B.1.1.529 (Omicron) variant, which partially escapes the majority of existing SARS-CoV-2 neutralizing antibodies [3,4] and has been reported to increase the number of hospitalizations among vaccinated adults [5]. Several reports have determined the ability of SARS-CoV-2 vaccines to generate immunity in HD patients and recommend implementing booster doses from highest to lowest priority-use groups [6,7,8]. 

Previous studies demonstrated a substantial increase in the antibody levels of naïve and COVID-19-recovered HD patients shortly after the second and third vaccine dose [9,10]. Others investigated the dynamics of post-vaccination antibody and T-cell responses for up to two months to determine the most appropriate timing for delivery of a booster dose. Results demonstrated comparable levels of total RBD antibodies and T-cells fifteen days and three months after the second vaccine dose between HD and HV [11]. This research group also investigated the immune response in HD patients, 90% without previous infection, and observed a booster effect on anti-RBD and neutralizing antibodies to different variants and a significant increase in SARS-CoV-2-S-IFN-γ-producing T-cells 46 days after receiving the third homologous mRNA vaccine dose [12]. More recent studies compared the immune response of non-infected naïve HD patients, who received four vaccine doses, with COVID-19-recovered HD patients, who only received three doses of the SARS-CoV-2 mRNA vaccine. The results indicated that, while there were no differences in the production of the proinflammatory cytokines interleukin-2 (IL-2) and tumor necrosis factor (TNF) by T-cells, better humoral immunity was observed in the convalescent-vaccinated compared to vaccinated-only HD patients [13]. These results suggest that the cellular and humoral immune responses provide different information regarding the immunological status of vaccinated HD patients that do not necessarily correlate with each other. Table 1 summarizes some of the most relevant and related studies.

Here, we monitored the long-term effects of SARS-CoV-2 vaccination in both, the cellular and humoral immune response in HD patients. Specifically, we evaluated the production of IFN-γ and IL-2 in the whole blood after stimulation with SARS-CoV-2 peptide pools and the IgG directed against Spike glycoprotein in HD patients and Healthy Volunteers (HV) with (COVID-19 recovered individuals) or without (naïve individuals) previous infection of SARS-CoV-2. Our results indicate that both naïve and COVID-19-recovered HD patients mount cellular and humoral immune responses comparable with HV individuals after the second and third dose of the 1273-mRNA SARS-CoV-2 vaccine. 

## 2. Results

We first monitored the cellular immune response in naïve subjects without previous SARS-CoV-2 infection by evaluating the production of IFN-γ in the whole blood after spike-specific peptide pool stimulation. Comparing the production of IFN-γ between HD patients and HV individuals, we observed a similar IFN-γ production between these two groups at both time points, V2+3M (*p* = 0.35) and V3+3M (*p* = 0.73) (Figure 1A). However, when comparing the production of IFN-γ between the two time points, we observed a significant decrease at V3+3M in HV individuals (*p* = 0.008). This suggests that the durability of cellular immunity decreases more rapidly in healthy individuals without previous SARS-CoV-2 infection after a booster dose.

We next measured the production of IFN-γ in COVID-19-recovered subjects. Comparing the production of IFN-γ between HD patients and HV individuals, we observed that, while similar amounts of IFN-γ were produced between these groups at V2+3M (*p* = 0.69), there was a significant IFN-γ decrease in HV individuals compared to HD patients at V3+3M (*p* = 0.003) (Figure 1B). When comparing the production of IFN-γ between the two time points, we also observed a significant decrease at V3+3M in HV individuals (*p* = 0.001). This suggests that the durability of cellular immunity is maintained in HD patients with a previous SARS-CoV-2 infection after a booster dose.

Next, we measured the production of IL-2 in naïve subjects. Comparing the production of IL-2 between HD patients and HV individuals, we observed a similar IL-2 production between these two groups at both time points, V2+3M and V3+3M (*p* = 0.33). However, when comparing the production of IL-2 between the two time points, we observed a significant decrease at V3+3M in HV individuals (*p* = 0.011) (Figure 1C). These results are consistent with data from Figure 1A, suggesting a decrease in the durability of the cellular immunity in healthy individuals without a previous SARS-CoV-2 infection after a booster dose. 

Finally, we compared the production of IL-2 in COVID-19-recovered subjects. Comparing the production of IL-2 between HD patients and HV individuals, we observed that, while similar amounts of IL-2 were produced between these groups at V2+3M (*p* = 0.15), there was a significant IL-2 decrease in HV individuals compared to HD patients at V3+3M (*p* = 0.0008) (Figure 1D). When comparing the production of IL-2 between the two time points, we also observed a significant decrease at V3+3M in HV individuals (*p* ≤ 0.0001). Overall, the cellular immunity results indicate that HD patients are able to mount and maintain a robust cellular immune response over time, while HV individuals decrease their ability to secrete both IFN-γ and IL-2 after a booster dose. 

We also monitored the humoral immune response in naïve subjects without previous SARS-CoV-2 infection by measuring IgG binding antibody units specific against the Spike glycoprotein (Figure 1E,F). Comparing the IgG levels between naïve HD patients and HV individuals, we observed a similar antibody production between these two groups at both time points, V2+3M (*p* = 0.43) and V3+3M (*p* = 0.72) (Figure 1E). However, when comparing the IgG levels at the two time points, we observed a significant increase in antibody production from V2+3M to V3+3M in both HD patients (*p* = 0.045) and HV individuals (*p* = 0.002). This suggests that booster doses significantly increase the cumulative antibody responses after repeated vaccinations. Comparing the IgG levels between previously infected HD patients and HV individuals, we observed that HD patients show significantly higher IgG levels compared to HV individuals at V2+3M (*p* = 0.009). However, these differences were not significant at V3+3M between both groups (*p* = 0.63), indicating that both COVID-19-recovered HD patients and HV individuals maintain their humoral response long-term after boosting (Figure 1F). 

## 3. Discussion

In this study, we examined the effects of SARS-CoV-2 vaccination on the humoral and cellular specific immune responses in HD patients compared to HV individuals with (COVID-19-recovered) or without (naïve) previous SARS-CoV-2 infection, three months after the second (V2+3M) and after the third (V3+3M) vaccination dose. Our results indicate that both naïve and COVID-19-recovered HD patients maintain strong cellular and humoral immune responses after receiving a third dose (booster), which is comparable or higher (significant increased at V3+3M for IFN-γ and IL-2) to HV individuals. 

Several studies have described that most HD patients can mount competitive immune responses [7,14,15]. Considering the humoral immune response alone, a recent cohort study reported the induction of robust and durable humoral immune response three months after receiving the BNT162b2 vaccine in naïve HD patients, following a two-dose immunization scheme [10]. Previous studies from David Navarro’s laboratory evaluated both the T-cell and Spike-specific reactive antibody responses in HD patients fifteen days and three months after two doses of mRNA vaccines (mRNA-123 and BNT162b2). In line with our results, they observed that HD patients develop SARS-CoV-2 antibody responses comparable to healthy controls (HC) (i.e., 95% rate of HD patient responders at 3M vs. 100% of HC responders at 3M) [9]. In addition, no differences between CD4+ or CD8+ T-cell responses were observed across groups, although we reported higher IFN-γ and IL-2 production in HD patients with previous SARS-CoV-2 infection compared to controls at V3+3M. It is likely that differences across studies regarding the clinical characteristics of patients, the time points under study, and the methodological approaches to evaluate T-cell immunity may, in part, explain the discrepancy. More recent data from Navarro’s laboratory confirmed the ability of HD patients to produce high levels of IgG production 46 days after the booster (anti-RBD antibodies were detected in 39/40 HD patients). Furthermore, SARS-CoV-2 specific-IFN-γ-producing CD8+ and CD4+ T-cell responses were detected in 35 and 36/37 of HD patients, respectively, indicating that mRNA COVID-19 vaccines induce a booster effect on both humoral and cellular immune responses in this immunocompromised group [12]. Similarly, Anft and colleagues recently described a stable cellular immunity with no differences in the production of proinflammatory cytokines (IL-2 and TNF) between four times vaccinated, non-infected HD patients compared to three times vaccinated, infected HD patients. However, a significant fade of neutralizing antibodies after SARS-CoV-2 vaccination in naïve HD patients (25%) compared to COVID-19-recovered HD patients (62.5%) was observed [13]. These results indicate significant differences between the humoral and cellular immune responses and highlight the importance of measuring both arms of the immune response in HD patients. Our results are consistent with these studies that report potent humoral and cellular immune responses in HD patients but further extend those findings, comparing HD data with HV and differentiating between patients with/without previous SARS-CoV-2 infection. 

A limitation of our study is the small sample size and the differences in vaccines between groups; HD patients were vaccinated with mRNA-1273 (Moderna), while HV individuals were vaccinated with BNT162b2 (Pfizer). Several studies have described that BNT162b2 vaccination induces diminished seroconversion compared to mRNA-1273 vaccination [7,16,17]. Nevertheless, the absolute indicators of the cellular and humoral immunity in HD and HV are comparable in our study, as we used the same methodological approaches to obtain the data. 

We conclude that HD patients develop potent cellular and humoral immune responses after COVID-19 vaccination over time, which is critical to lower the rate of COVID-19-related hospitalizations in this vulnerable group of patients [18]. While the precise mechanisms behind the robust immune response induced by SARS-CoV-2 vaccination, we hypnotize that trained immunity, which has previously been associated with COVID-19 vaccination and infection [19,20], may be responsible, in part, to the delicate balance between the protective and the inflammatory state of HD patients [21]. Although further studies are required to demonstrate the relationship between protection and specific T-cell or serological immune responses, the development of strong cellular and humoral immune responses reported here may help guide future vaccination strategies in immunocompromised groups of patients. 

## 4. Materials and Methods

### 4.1. Experimental Design

In this study, peripheral blood from HD patients was drawn before hemodialysis (*n* = 38), while in HV individuals (*n* = 30) it was prospectively collected. The second vaccination dose of HD patients and Healthy Volunteers occurred in May 2021. The third vaccination dose of HD patients and Healthy Volunteers occurred in October 2021. All blood extractions were performed approximately 90 days after second vaccination dose and 90 days after third vaccination dose. All individuals were based in the Comunidad de Madrid, Spain. Healthy volunteers were obtained from Hospital Universitario La Paz in Madrid and HD patients were obtained from Hospital Universitario Puerta de Hierro in Madrid. Blood extractions from HD patients [Naïve (*n* = 19), COVID-19-recovered patients (*n* = 19)], HV individuals [naïve (*n* = 15), and COVID-19-recovered HV (*n* = 15)] were collected three months after the second (V2+3M) and three months after the third vaccine dose (V3+3M). Table 2 and Table 3 summarize HD patient and HV individuals’ characteristics. 

### 4.2. SARS-CoV-2 Peptide Pools and Whole-Blood Culture Assays

Lithium-heparinized blood samples were collected before the start of dialysis. On the same day, 320 µL of whole blood was mixed with 80µL of RPMI and stimulated with PepTivator ^®^ SARS-CoV-2 Peptide Pools (S; 2 µg/mL, M; 2 µg/mL) or a DMSO control. After 16–20 h of culture, supernatant (plasma) was collected and stored at −20 °C for further cytokine quantification, as previously reported [14]. For previous SARS-CoV-2 infection detection, whole blood cultures were incubated with a peptide pool against SARS-CoV-2 membrane (M) protein (2 μg/mL). 

### 4.3. Spike-Specific IgG Quantification and Analysis

To study the specific serologic response against SARS-CoV-2, plasma from HD patients and HV was collected. The Liaison ^®^ SARS-CoV-2 TrimetricS IgG assay (Diasorin, Stillwater, MN, USA) was used for semiquantitative detection of IgG directed against the Spike glycoprotein. Values over 33.8 BAU/mL were considered positive. 

### 4.4. Cytokine Quantification and Analysis

Cytokine concentrations in the supernatants (plasma) were quantified using ELLA with microfluidic multiplex cartridges measuring IFN-γ and IL-2 release following the manufacturer’s instructions (ProteinSimple, San Jose, CA, USA). The cytokine levels present in plasma stimulated with DMSO were subtracted from the corresponding Peptide-pool stimulated samples, as previously reported [22].

### 4.5. Statistics

Statistical analyses were performed by Two-Way ANOVA and Šídák’s multiple comparison tests. Normality of data was tested using D’Agostino and Pearson tests for normal distribution. Paired t test and unpaired t test were also used as appropriate, using Graphpad PRISM 9.01 (Graphpad Software, La Jolla, CA, USA). 

## Figures and Tables

**Figure 1 pharmaceuticals-16-00574-f001:**
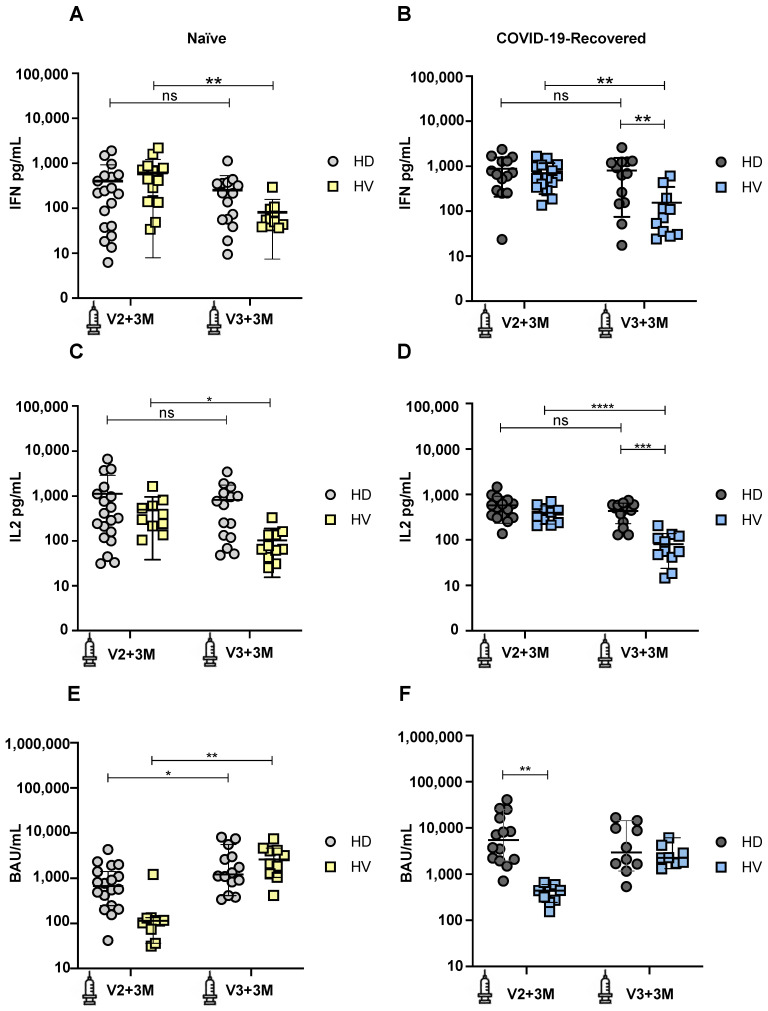
Development of cellular and humoral immune responses after SARS-CoV-2 vaccination in COVID-19-recovered and naïve hemodialysis (HD) patients and healthy volunteers (HV) 3 months after second (V2+3M) and 3 months after third (V3+3M) vaccination dose. (**A**) IFN-γ production in naïve HD patients (light grey symbols) and HV individuals (yellow symbols) at V2+3M and V3+3M. (**B**) IFN-γ production in COVID-19-recovered HD patients (dark grey symbols) and HV individuals (blue symbols) at V2+3M and V3+3M. (**C**) IL-2 production in naïve HD patients and HV individuals at V2+3M and V3+3M (**D**). IL-2 production in COVID-19-recovered HD patients and HV individuals at V2+3M and V3+3M. (**E**) SARS-CoV-2 spike-specific IgG serum levels in naïve HD patients and HV individuals at V2+3M and V3+3M. (**F**) Comparison of SARS-CoV-2 spike-specific IgG binding antibody units (BAU) in COVID-19-recovered HD patients and HV individuals at V2+3M and V3+3M. Values higher than 33.8 BAU/mL were considered positive. <0.05 (*), <0.005 (**), <0.0005 (***), and <0.0001 (****). Data are shown as mean ± SEM.

**Table 1 pharmaceuticals-16-00574-t001:** Summary of studies evaluating the humoral and cellular immune response in HD patients.

Article	Studied Type of Response	Follow Up	Type of Patients	Vaccine Type	Outcome
Paal M. (2021) [6]	Humoral	3–6 weeks after V2	Control and HD patients; Naïve and COVID-19-recovered	mRNA vaccines ^1^	Control individuals had significantly higher Ab titers compared to HD patients
Stumpf J. (2021) [7]	Humoral and cellular	Baseline, 3–4 weeks after V1, 4–5 weeks after V2	Control and HD patients; Naïve.	mRNA vaccines ^1^	HD patients present a higher seroconversion rate compared to similar tested medical personnel
Bensouna I. (2021) [9]	Humoral	After V2 and 3 weeks after V3	HD patients; Naïve and COVID-19-recovered	BNT162b2	V3 substantially increased Ab titers in HD patients compared to V2
Panizo N. (2022) [11]	Humoral and cellular	Baseline, Day 15, and 3 months after V2	Control and HD patients	mRNA vaccines ^1^	HD patients develop similar humoral response compared to controls. No differences were found in cellular immune responses
Panizo N. (2022) [12]	Humoral and cellular	46 days after V3	Control and HD patients	mRNA vaccines ^1^	Boosted humoral and cellular responses in HD patients
Mirioglu S. (2023) [10]	Humoral	1 and 3 months after V2	HD patients; Naïve	BNT162b2 and Coronavac	HD patients had induced humoral response after booster dose
Anft M. (2023) [13]	Humoral and cellular	158 days after V4	HD patients; Naïve and COVID-19-recovered	mRNA vaccine	HD patients present high humoral and cellular responses

^1^ mRNA- 1273 and BNT162b2, V1: First vaccine dose, V2: Second vaccine dose, V3: Third vaccine dose, V4: Fourth vaccine dose, Ab: Antibody.

**Table 2 pharmaceuticals-16-00574-t002:** Naïve and COVID-19 recovered HD patients’ characteristics.

Characteristics	Naïve HD * Patients *N* = 19	N (Partial)	COVID-19 HD * Patients *N* = 19	N (Partial)
Male gender	47.4%	9	55.0%	10
Age, years (Mean ± SD)	64.0 ± 12.4	-	65.3 ± 13.2	-
Active smoking	15.8%	3	15.0%	3
HD vintage, months (Mean ± SD)	96.1 ± 102.5	-	81.4 ± 72.2	-
Use of EPO *	89.4%	17	100.0%	19
Previous kidney transplantation	26.3%	5	35.0%	7
Comorbidities	-	-	-	-
Obesity	15.8%	3	30.0%	6
Hypertension	89.5%	17	95.0%	18
Diabetes mellitus	31.6%	6	45.0%	9
Ischemic heart disease	31.6%	6	15.0%	3
Dyslipidemia	68.4%	13	60.0%	12
Cause of end-stage renal disease				
Diabetic nephropathy	28.6%	2	40.0%	8
Hypertensive nephrosclerosis	5.3%	1	10.0%	2
Glomerulonephritis	5.3%	1	15.0%	3

* HD: Hemodialysis, EPO: recombinant Erythropoietin treatments.

**Table 3 pharmaceuticals-16-00574-t003:** Naïve and COVID-19 recovered HV individuals’ characteristics.

Characteristics	Naïve HV *N* = 15	N (Partial)	COVID-19 HV *N* = 15	N (Partial)
Male gender	13.33%	2	6.66%	1
Age, years (Mean ± SD)	46 ± 14.38	-	46 ± 16.98	-
Active smoking	13.33%	2	13.33%	2
Comorbidities	-	-	-	-
Obesity	13.33%	2	13.33%	2
Hypertension	40%	6	46.66%	7
Diabetes mellitus	6.66%	1	13.33%	2
Ischemic heart disease	0%	0	0%	0
Dyslipidemia	46.66%	7	33.33%	5

## Data Availability

The original data contribution presented in the study are included in the article, further inquiries can be directed to the corresponding authors.

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
