# Peer review of "Maintenance of Potent Cellular and Humoral Immune Responses in Long-Term Hemodialysis Patients after 1273-mRNA SARS-CoV-2 Vaccination"

_pharmaceuticals, 2023, doi:10.3390/ph16040574_

Round 1
Reviewer 1 Report
In this interesting and clearely written prospective study, authors analyze last immunological responses both humoral and cellular in hemodialysis patients and heathy controls 3 months after their second and their third m-RNA vaccine dose. They demonstrate an equal response between HD and healthy controls and the lack of waning of anti-spike antibodies together with a decrease of interferon gamma and IL2 driven by vaccine peptides after the third dose.
2 minor comments:
Authors compare results with two-way ANOVA followed with Sidak multiple comparison test.
I guess that authors have confirm previously the gaussian distribution of their values with D'Agostino-Pearson test. Authors , please confirm and add this important point in the statistical analysis paragraph since number of patients in the HD groups as healthy volunteers ere low numbers (and normality test is required in such cases)
To compare values obtained in this study with those previously published , a table summarizing the main results of immunological studies would be useful.
Author Response
1) Statistical analyses were performed by Two-Way ANOVA and Šídák’s multiple comparisons test. Normally of data was tested using D’Agostino and Pearson test for normal distribution. Paired t test and unpaired t test was also used as appropriate, using Graphpad PRISM 9.01 (Graphpad Software, La Jolla, CA). This is now included in the manuscript.
2) A table summarizing main results of immunological studies has been added as Table 2.
Reviewer 2 Report
The authors investigated cellular and humoral immune responses in patients undergoing hemodialysis after SARS-CoV2 vaccination.
Although there have been numerous papers in terms of the humoral response after SARS-CoV vaccination, this study assesses IFN-γ and IL-2 secretion levels.
The topic of this study is important. However, there are some critical drawbacks to this study.
The authors compared the patients undergoing hemodialysis with healthy volunteers in this study.
Figure 1 showed that patients undergoing hemodialysis showed higher humoral and cellular responses after the third dose.
This may be attributed to not the patient background but to the difference in vaccinations.
In addition, the authors did not show the background of healthy volunteers. It is mandatory to show the details of control.
Furthermore, the study period and the place where the participants were included were not shown in the manuscript.
The period between the second dose and the third dose of vaccinations was not shown, either.
As the authors acknowledged, the small number of participants in this study is one of the important issues. How did the authors estimate the sample size in this study?
In my opinion, these problems above should hinder obtaining an accurate conclusion.
Author Response
- Background of healthy volunteers is included table 2.
- Study period and place has been included in the Experimental design section.
- Period between the second dose and the third dose of vaccinations has been included in the Experimental design section.
- We estimated the sample size of this study using a paired measurements in two groups. The estimated sample size to recognize a statistically significant difference greater or equal to 1 unit was 32 individuals per group (HD patients vs HV individuals). This was stablished taking into consideration a two-sided test with an 0.05 alpha risk, a 0.20 beta risk and an estimated common standard deviation of 1. The correlation coefficient between the initial and final measurement was 0.2. An anticipated drop-out rate of 20% was also taken into consideration. However, our study was stablished during early phases of vaccination and the amount and availability of samples from HD patients was limited with time.
Reviewer 3 Report
The study is interesting, and the manuscript is well-written. The results confirmed the results of previous studies showing an adequate cellular and humoral immune response after mRNA vaccination against SARS-CoV2 but also extended the above knowledge to different time points.
I have a minor comment: Considering that the HD population is immunocompromised and the rate or the sustainability of postvaccination immune response is diminished, a comment or even a suggestion about the possible mechanism that leads to an adequate and sustainable immune response to the SARS-CoV2 vaccine would be well-come.
Author Response
We have added a comment in the discussion section hypothesizing that trained immunity may be responsible for the sustainable immune response in HD patients.
Round 2
Reviewer 2 Report
The authors addressed my concerns.